# Regulation of alternative polyadenylation by Nkx2-5 and Xrn2 during mouse heart development

Keisuke Nimura[1]*, Masamichi Yamamoto[2], Makiko Takeichi[1], Kotaro Saga[1], Katsuyoshi Takaoka[3], Norihiko Kawamura[1,4], Hirohisa Nitta[1], Hiromichi Nagano[1], Saki Ishino[5], Tatsuya Tanaka[5], Robert J Schwartz[6], Hiroyuki Aburatani[7], Yasufumi Kaneda[1]*

[1]Division of Gene Therapy Science, Osaka University Graduate School of Medicine, Suita, Japan; [2]Department of Nephrology, Kyoto University Graduate School of Medicine, Kyoto, Japan; [3]Developmental Genetics Group, Graduate School of Frontier Biosciences, Osaka University, Suita, Japan; [4]Department of Urology, Osaka University Graduate School of Medicine, Suita, Japan; [5]Center for Medical Research and Education, Osaka University Graduate School of Medicine, Suita, Japan; [6]Department of Biology and Biochemistry, University of Houston, Houston, Unites States; [7]Genome Science Division, Research Center for Advanced Science and Technology (RCAST), The University of Tokyo, Tokyo, Japan

*For correspondence: nimura@ gts.med.osaka-u.ac.jp (KN); kaneday@gts.med.osaka-u.ac.jp (YK)

**Competing interests:** The authors declare that no competing interests exist.

**Abstract** Transcription factors organize gene expression profiles by regulating promoter activity. However, the role of transcription factors after transcription initiation is poorly understood. Here, we show that the homeoprotein Nkx2-5 and the 5'-3' exonuclease Xrn2 are involved in the regulation of alternative polyadenylation (APA) during mouse heart development. Nkx2-5 occupied not only the transcription start sites (TSSs) but also the downstream regions of genes, serving to connect these regions in primary embryonic cardiomyocytes (eCMs). Nkx2-5 deficiency affected Xrn2 binding to target loci and resulted in increases in RNA polymerase II (RNAPII) occupancy and in the expression of mRNAs with long 3'untranslated regions (3' UTRs) from genes related to heart development. siRNA-mediated suppression of Nkx2-5 and Xrn2 led to heart looping anomaly. Moreover, $Nkx2-5$ genetically interacts with $Xrn2$ because $Nkx2-5^{+/-}Xrn2^{+/-}$, but neither $Nkx2-5^{+/-}$ nor $Xrn2^{+/-}$, newborns exhibited a defect in ventricular septum formation, suggesting that the association between Nkx2-5 and Xrn2 is essential for heart development. Our results indicate that Nkx2-5 regulates not only the initiation but also the usage of poly(A) sites during heart development. Our findings suggest that tissue-specific transcription factors is involved in the regulation of APA.

## Introduction

Transcription factors and chromatin regulators orchestrate the processes of heart development by positively and negatively regulating thousands of genes (*Bruneau, 2010*; *Nimura et al., 2009*; *Prall et al., 2007*; *Srivastava, 2006*; *Takeuchi and Bruneau, 2009*; *Takeuchi et al., 2011*). Mutations in transcription factors and chromatin regulators cause congenital heart disease (CHD) by disrupting gene expression profiles that are tightly regulated by transcription factor networks (*Bruneau, 2008*; *Srivastava, 2006*). However, the mechanisms by which transcription factor deficiencies cause CHD are not fully understood.

**eLife digest** About one in every hundred babies is born with problems that either affect the structure of the heart or how it works. These problems are known as congenital heart disease, and result when the development of the heart is disrupted. How the heart develops is determined by thousands of genes whose activity or "expression" must be precisely regulated. Proteins called transcription factors can control gene expression; therefore, researchers may discover new ways of treating congenital heart disease if they can understand how transcription factors work during normal heart development.

To produce a protein, the information in a gene must first be "transcribed" to form a molecule of messenger RNA (mRNA). Not all of the mRNA sequence is subsequently "translated" to form the protein; this includes a stretch at the end of the mRNA called the 3' untranslated region. The length of the 3' untranslated region for a particular mRNA may vary depending on the type of cell it has been produced in, and this length can influence how efficiently the mRNA is translated to form a protein. However, it was not clear what changes the length of the 3' untranslated region.

Nimura et al. have now studied mice to investigate the role of a transcription factor called Nkx2-5, which was known to be important for heart development. This revealed that in addition to its expected role in starting the transcription of genes that are important for heart development, Nkx2-5 also controls the length of 3' untranslated regions of certain mRNAs. To do so, Nkx2-5 binds to a protein called Xrn2 that stops transcription when the end of the gene is reached.

Mouse embryos that lacked Nkx2-5 produced mRNAs containing long 3' untranslated regions from genes related to the development of the heart. Furthermore, suppressing the activity of both Nkx2-5 and Xrn2 resulted in the embryos developing heart defects.

The findings of Nimura et al. suggest that transcription factors found in specific tissues are responsible for the different lengths of 3' untranslated regions in mRNAs in different tissues. Furthermore, incorrectly regulating the length of these regions appears to be linked to the development of congenital heart disease. The next step is to understand exactly how the failure to correctly regulate the length of 3' untranslated regions contributes to congenital heart disease.

Nkx2-5, Gata4, and Tbx5 are key cardiac transcription factors that coordinate transcription networks during heart development (*Akazawa and Komuro, 2005*; *Stennard and Harvey, 2005*). Haploinsufficiency of these genes can cause CHD in humans, and mice lacking any of these transcription factors exhibit severe defects in heart development (*Akazawa and Komuro, 2005*; *Stennard and Harvey, 2005*). Recent genome-wide analyses of transcription factors, including Nkx2-5, Gata4, and Tbx5, have revealed that these transcription factors assist in the formation of active enhancers in the HL1 cardiomyocyte cell line and in murine adult hearts (*He et al., 2011*; *Schlesinger et al., 2011*; *van den Boogaard et al., 2012*). He *et al.* demonstrated that multiple transcription factors activate certain cardiac enhancers without p300 (*He et al., 2011*). In *Drosophila*, cardiac transcription factors converge on heart enhancer regions; however, the collective binding of these transcription factors to the enhancer regions does not require a conserved DNA motif (*Junion et al., 2012*). Although the regulation of enhancer activity by cardiac transcription factors has been extensively studied, the other roles of these transcription factors during heart development remain poorly understood.

Transcription factors require several chromatin regulators to precisely regulate gene expression. Indeed, the results from several histone methyltransferase-knockout mouse studies and genetic studies of human CHD patients have revealed the importance of histone modifications during heart development (*Delgado-Olguín et al., 2012*; *Nimura et al., 2009*). In particular, Nkx2-5 regulates gene expression in conjunction with Whsc1 (Wolf-Hirschhorn Syndrome 1, also known as NSD2 or MMSET), a histone H3 lysine 36 (H3K36) methyltransferase. H3K36 methylation is associated with transcribed genomic regions and has several roles in transcription, including transcriptional repression, alternative RNA splicing, and DNA mismatch repair (*Li et al., 2013*; *Luco et al., 2010*). Therefore, Nkx2-5 may be involved in both transcriptional activation and the subsequent events. However, it is not clear whether particular transcription factors regulate events that occur after transcription initiation during heart development.

In this study, we determined the genome-wide occupancy of transcription factors that are critical for heart development and the factors associated with these transcription factors as well as the histone modification signatures in embryonic hearts. The results indicate a role for transcription factors in alternative polyadenylation (APA). Furthermore, we discovered that Nkx2-5 is associated with both transcription start sites (TSSs) and downstream regions of genes and that Nkx2-5 controls APA in conjunction with the 5'-3' exonuclease Xrn2. Simultaneously suppressing Nkx2-5 and Xrn2 caused heart-looping abnormalities. Moreover, *Nkx2-5* genetically interacted with *Xrn2* during heart development. Our findings suggest that Nkx2-5 is involved in the regulation of the length of the 3' UTR and may help to elucidate the mechanisms by which transcription factor deficiencies can cause diseases such as CHD.

## Results

### Nkx2-5 deficiency increases transcription from regions downstream of transcription termination sites

To elucidate how Nkx2-5, Gata4, and Tbx5 regulate transcription, we examined the genomic target regions of these transcription factors and the transcription factor-associated factors as well as the chromatin status in mouse E12.5 hearts using chromatin immunoprecipitation-sequencing (ChIP-seq) (*Figures 1A,B*, *Figure 1—figure supplement 1*, *Figure 1—figure supplement 2*, *Figure 1—figure supplement 3*, *Figure 1—source data 1*, and *Figure 1—figure supplement 4*) (*Li et al., 2007*; *Lickert et al., 2004*; *Nimura et al., 2009*). Embryonic hearts at this stage are primarily (>90%) composed of cardiomyocytes that do not express Thy1 (fibroblasts, T-lymphocytes, and neuronal markers) (*Ieda et al., 2009*). Defects in the ventricular septum and atrial septum, which are most frequently found in CHDs (*Feng et al., 2002*), form during this stage (*Henderson and Copp, 1998*). Nkx2-5 and Tbx5 co-occupied the same global genomic regions as RNA polymerase II (RNAPII) and the heterogeneous nuclear ribonucleoprotein Raver1 (*Figure 1A*). Nkx2-5 and Tbx5 were found to associate with the poised serine 5-phosphorylated form of RNAPII (RNAPII-S5P) (*Kuehner et al., 2011*), although a physical association between Nkx2-5 and Tbx5 was not detected (*Figure 1B*). In contrast, Gata4 was found in a different cluster (*Figure 1A*). We next examined whether the occupancy of Nkx2-5 and Tbx5 is correlated with gene expression levels because these two transcription factors are associated with RNAPII. Nkx2-5 and Tbx5 were significantly enriched around the TSSs and downstream regions of genes that were highly expressed (*Figures 1C*, *Figure 1—figure supplement 5*, *Figure 1—figure supplement 6*, and *Figure 1—figure supplement 7*). These results suggest the possibility that Nkx2-5 and Tbx5 may be involved not only in enhancer activity regulation (*He et al., 2011*; *Schlesinger et al., 2011*; *van den Boogaard et al., 2012*) but also in post-transcriptional mRNA processing. To elucidate whether Nkx2-5 and Tbx5 play a role in regulating 3'-end processing, we examined changes in poly(A)-tailed mRNA in transcription factor-knockdown embryonic cardiomyocytes (eCMs) using mRNA-seq. Nkx2-5 knockdown increased the expression of long 3' UTRs in *Tnnt2* (Troponin T2, cardiac) and *Atp2a2* (ATPase, Ca++ transporting, cardiac muscle, slow twitch 2) transcripts (*Figures 1D*, *Figure 1—figure supplement 5*, and *Figure 1—figure supplement 8*). We also detected increased expression of long 3' UTRs in *Nkx2-5*-knockout E9.5 hearts (*Figures 1E* and *Figure 1—figure supplement 9*). These results suggest a role for Nkx2-5 in the regulation of APA.

### A link between Nkx2-5-dependent chromatin looping and termination of RNA polymerase II activity

Recent studies have revealed that enhancer-promoter looping mediated by transcription factors is important for gene regulation (*Kagey et al., 2010*; *Wang et al., 2011*). As shown in *Figure 1*, Nkx2-5 is located at the TSSs and downstream regions of highly expressed genes and is involved in regulating 3'-end processing, which implies that Nkx2-5 may organize chromatin conformations between TSSs and downstream regions of genes and that this Nkx2-5-mediated chromatin conformation may be related to 3'-end processing. A chromatin conformation capture (3C) assay demonstrated that TSSs interact with the downstream regions of two genes, *Tnnt2* and *Atp2a2*, which are highly expressed in eCMs. The looping between the TSSs and downstream regions of the genes (TSS-downstream looping) was dependent on Nkx2-5 but not on Gata4 or Tbx5 (*Figure 2A* and

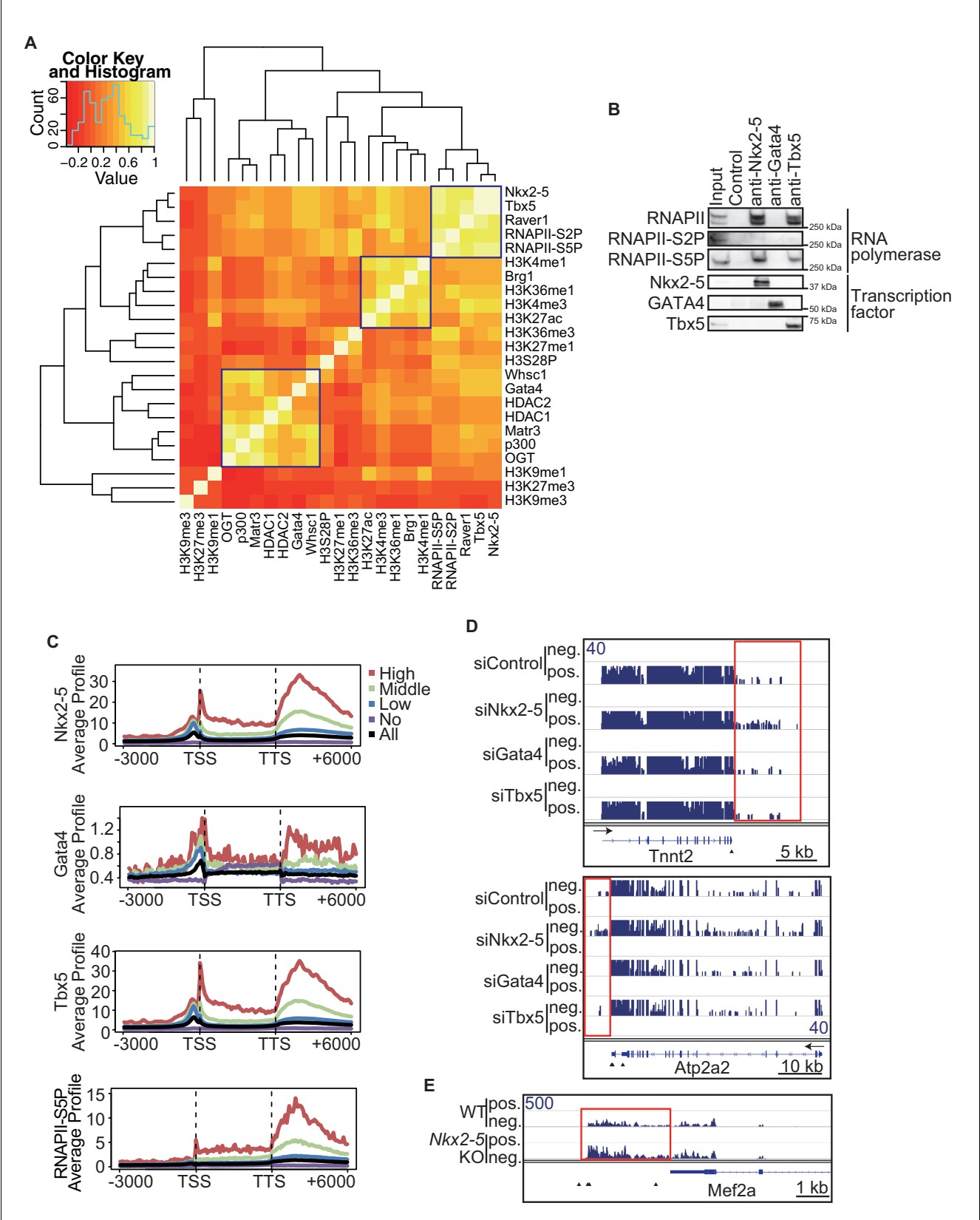

**Figure 1.** Nkx2-5 deficiency increases transcription from regions downstream of transcription termination sites. (**A**) Co-occupancies of each pair of factors and histone modifications are shown. White indicates a high correlation, and red indicates a low correlation. (**B**) Nkx2-5, Tbx5, and Gata4 were

*Figure 1 continued on next page*

*Figure 1 continued*

immunoprecipitated from nuclear extracts of E12.5 hearts with the indicated antibodies. Co-immunoprecipitates and aliquots (6%) of the input proteins were analyzed by Western blotting with the indicated antibodies. (C) Average ChIP-seq signal profiles over a 3-kb meta-gene, including 3 kb upstream and 3 kb downstream. The lines correspond to genes with High, Middle, Low, and No expression and all RefSeq genes. (D and E) Genome browser representation of strand-specific RNA-seq tag counts from eCMs transfected with the indicated siRNAs (D) and E9.5 *Nkx2-5⁻/⁻* hearts (E). The red boxes indicate read-through RNAs. neg., negative strand; pos., positive strand. The arrow heads show polyadenylation sites.

The following source data and figure supplements are available for figure 1:

**Source data 1.** Overlap of peaks between transcription factors and between the results from this study and those from previously published studies.

**Figure supplement 1.** Transcription factors-associated proteins in E12.5 hearts.

**Figure supplement 2.** ChIPseq replicate correlations.

**Figure supplement 3.** In vivo transcription factor binding motif by native ChIPseq.

**Figure supplement 4.** Validation of the antibodies used for ChIP-seq.

**Figure supplement 5.** Genome browser representation at Tnnt2 and Atps2a2 loci.

**Figure supplement 6.** Heatmap of factor occupancy and histone modification enrichment for 8 kb regions centred on TSSs (left panel) and TTSs (middle panel) are shown with reference to the RefSeq gene expression level (right panel).

**Figure supplement 7.** Average signal profiles over a 3 kb meta-gene including 3 kb upstream and 3 kb downstream.

**Figure supplement 8.** Genome browser representation of strand-specific RNA-seq tag counts from eCMs transfected with the indicated siRNAs.

**Figure supplement 9.** mRNA with long 3'UTR in Nkx2-5-knockout embryonic hearts.

*Figure 2—figure supplement 1*). However, Nkx2-5 knockdown did not affect the TSS-downstream looping of *Tnni1* and did not increase the expression of long 3' UTRs (*Figure 2—figure supplement 2*). Furthermore, the binding of Nkx2-5 to the TSSs and downstream regions of *Tnnt2* and *Atp2a2* in eCMs was also confirmed using ChIP-qPCR at the Nkx2-5-bound downstream regions (*Figure 2B*). This finding suggests that the loss of TSS-downstream looping is related to the increased expression of long 3' UTRs.

To investigate whether TSS-downstream looping is involved in transcription regulation, we examined whether the occupancy of the elongating serine 2-phosphorylated form of RNAPII (RNAPII-S2P) (*Kuehner et al., 2011*) was increased at these downstream regions in accordance with the increased expression of long 3' UTRs. RNAPII-S2P binding downstream of *Tnnt2* and *Atp2a2* was increased in Nkx2-5-knockdown eCMs (*Figures 2C and D*). Consistent with the increase in RNAPII at downstream regions, Nkx2-5 knockdown increased the expression of long 3' UTRs; however, the mRNA expression of the coding regions was not substantially increased (*Figures 2E,F*, and *Figure 2—figure supplement 3*). Although Gata4 knockdown may affect the regulation of APA, we could not detect a significant difference in the APA of *Tnnt2* and *Atp2a2* (*Figure 2F*). These findings further support the possibility that Nkx2-5-mediated TSS-downstream looping is associated with RNAPII termination.

To investigate whether mRNAs with long 3' UTRs in Nkx2-5-knockdown eCMs are enriched for specific functional annotations, we performed Gene Ontology (GO) analysis (*Huang et al., 2009a; 2009b*). We observed that the mRNAs with long 3' UTRs that were increased by the knockdown and knockout of Nkx2-5 were enriched for GO terms related to heart development (*Figure 2—figure supplement 4*), whereas in mRNAs with long 3' UTRs that were increased by the knockdown of Tbx5 and Gata4, none of GO terms were linked to heart development (*Figure 2—figure supplement 4*). These results suggest that Nkx2-5 plays essential roles in regulating the APA of genes related to heart development.

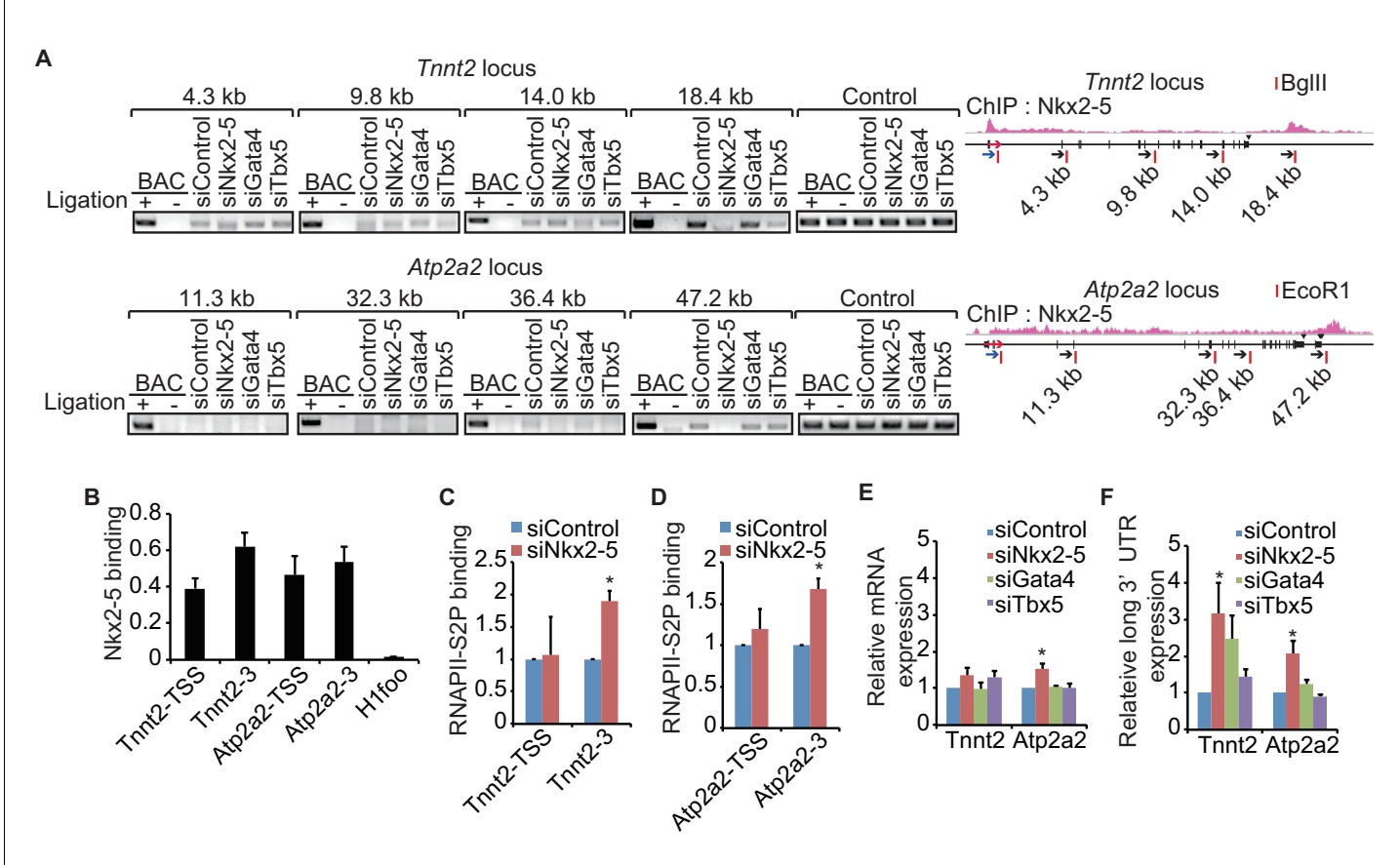

**Figure 2.** A link between Nkx2-5-dependent chromatin conformation and RNAPII. (**A**) Chromatin conformation capture (3C) analysis of the TSSs and downstream regions of *Tnnt2* and *Atp2a2* in the indicated siRNA-treated eCMs. The corresponding BACs for the regions were used as controls. Undigested regions at *Tnnt2* and *Atp2a2* were used as Controls. Red arrow, direction of transcription; blue arrow, anchoring primer; black arrow, primer; red line, restriction enzyme site. The arrow heads show polyadenylation sites. (**B–D**), Relative Nkx2-5 (**B**) and RNAPII-S2P occupancy (**C** and **D**) at the TSSs (-TSS) and downstream regions (-3) of *Tnnt2* (18.4 kb), *Atp2a2* (47.2 kb), and the silent *histone H1foo* gene, which served as a negative control, was analyzed by ChIP. (**E** and **F**) qRT-PCR analysis of mRNA expression from the coding region (**E**) and expression of long 3' UTRs (**F**) of *Tnnt2* and *Atp2a2*, normalized to *Rplp2*. Error bars indicate the mean ± s.e.m. (*n* = 3). *, *p* < 0.05.

The following source data and figure supplements are available for figure 2:

**Source data 1.** Source data for *Figure 2* and *Figure 2—figure supplement 1* and *3*.
**Figure supplement 1.** Quantification of 3C and western blotting data.
**Figure supplement 2.** The chromatin conformation of *Tnni1* is independent of Nkx2-5.
**Figure supplement 3.** siRNA knockdown efficiencies of three different siRNAs for each gene.
**Figure supplement 4.** Functional annotations of genes with increased and decreased mRNA with long 3'UTR.

## Nkx2-5 associates with the 5'-3' exonuclease Xrn2

To elucidate the molecular mechanisms by which Nkx2-5 regulates APA, we examined the transcription termination factors associated with Nkx2-5 in the embryonic heart. We could detect an association between Nkx2-5 and the transcription termination factor Xrn2, which has 5'-3' exonuclease activity (*Kuehner et al., 2011*) (*Figure 3A*). However, no association was detected between Nkx2-5 and the RNA helicases, Senataxin, Ddx5, and Dhx9 (*Figure 3A*), although it has been reported that

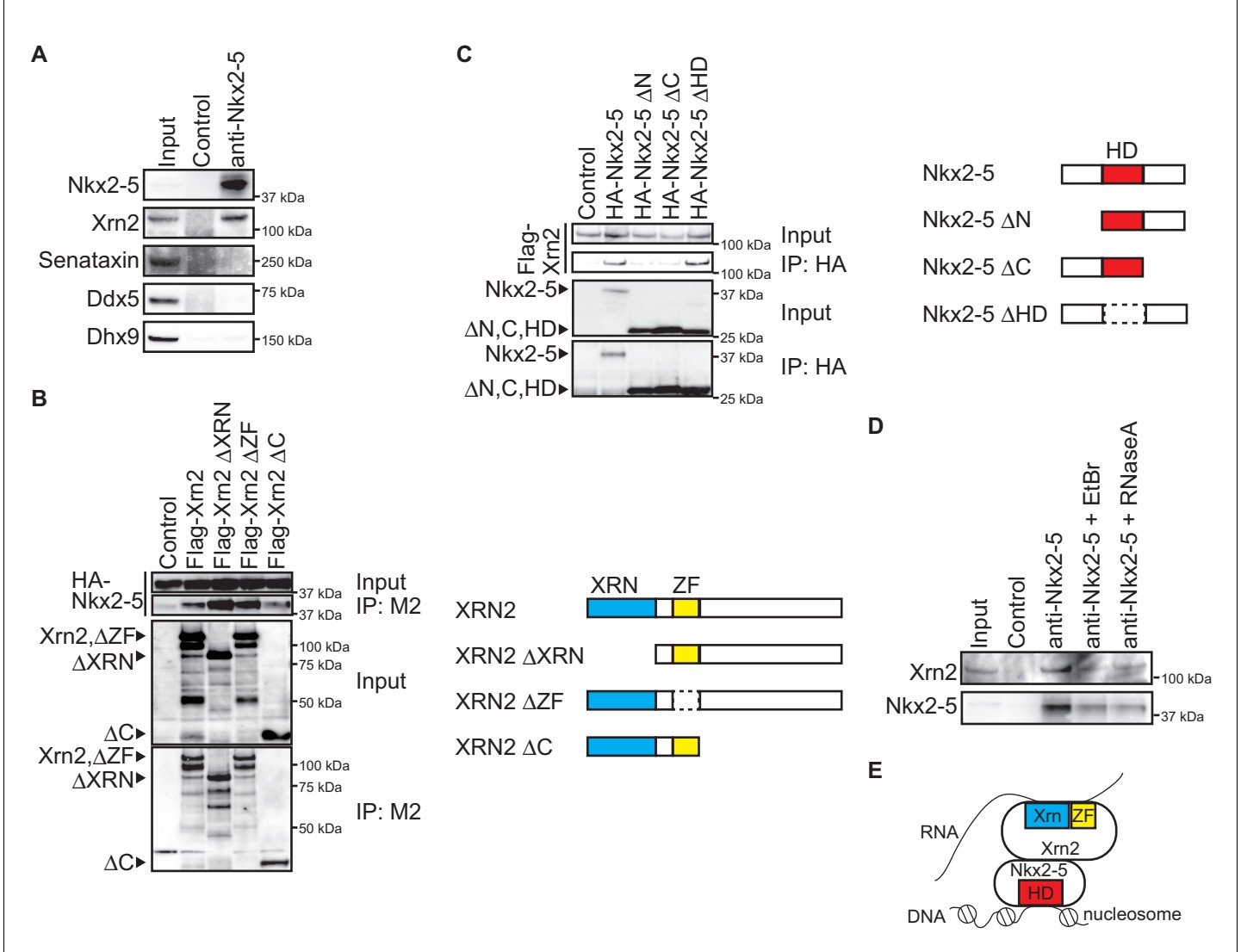

**Figure 3.** Nkx2-5 associates with the 5'-3' exonuclease Xrn2. (**A**) Co-immunoprecipitates derived using the indicated antibodies from nuclear extracts of E12.5 hearts and aliquots (6%) of the input proteins were analyzed by Western blotting. (**B**) Xrn2 and Nkx2-5 deletion mutants were transfected into C3H10T1/2 cells. Co-immunoprecipitates derived using the M2 antibody and aliquots (7%) of the input proteins were analyzed by Western blotting. Schematic presentation of Xrn2 and its deletion mutants is shown at the right panel. (**C**) Nkx2-5 and Xrn2 deleting mutants were transfected into C3H10T1/2 cells. Co-immunoprecipitates derived using the HA antibody and aliquots (7%) of the input proteins were analyzed by Western blotting. Schematic presentation of Nkx2-5 and its deletion mutants is shown at the right panel. (**D**) Co-immunoprecipitates derived using the indicated antibodies from nuclear extracts of E12.5 hearts exposed to 20 μg/ml EtBr or 50 μg/ml RNaseA as well as aliquots (6%) of the input proteins were analyzed by Western blotting. (**E**) Summary of interacting regions between Nkx2-5 and Xrn2. HD, homeodomain; Xrn, Xrn domain; ZF, zinc finger.

these RNA helicases play an essential role in processing the ends of transcripts and that this process is regulated by the circadian clock PERIOD complex (*Padmanabhan et al., 2012*). To confirm the association between Nkx2-5 and Xrn2, we examined regions that may be important for their association. We found that the C-terminus of Xrn2 was important due to its association with Nkx2-5 (*Figure 3B*). The N- and C-termini of Nkx2-5 were crucial due to their association with Xrn2, whereas the Nkx2-5 homeodomain (HD) was not required (*Figure 3C*). Next, we examined whether the association between Nkx2-5 and Xrn2 is dependent on DNA or RNA. Xrn2 co-immunoprecipitated with Nkx2-5 from eCM nuclear extracts that were treated with ethidium bromide (EtBr) to release DNA from proteins (*Heale et al., 2006*) (*Figure 3D*). Although this concentration of EtBr (20 μg/ml) reduced the association of Whsc1 with histone (*Nimura et al., 2009*), the association of Nkx2-5 with

Xrn2 was resistant to EtBr treatment. Furthermore, the association between these factors was also resistant to RNase A (50 µg/ml) treatment (*Calvo and Manley, 2001*). These data suggest that Nkx2-5 may be associated with Xrn2 in a DNA- or RNA-independent manner and that Nkx2-5 could recruit Xrn2 to target regions of the genome based on its ability to recognize these target regions with its HD (*Figure 3E*).

## Nkx2-5 functions together with Xrn2 to regulate APA

To determine whether Xrn2 regulates the APA of *Tnnt2* and *Atp2a2*, Xrn2 was repressed by transfection with a sequence-specific siRNA in eCMs (*Figures 4A,B, and C*). Xrn2 knockdown increased the expression of *Tnnt2* and *Atp2a2* transcripts with long 3'UTRs but did not alter transcription of the coding sequence of these genes (*Figures 4B and C*). The increase in expression of long 3'UTR of *Tnnt2* and *Atp2a2* was also detected in chromatin-fractioned RNA (*Figure 4—figure supplement 1*). Furthermore, Xrn2 expression was not affected by the knockdown of Nkx2-5, Gata4, or Tbx5 (*Figure 4D*). Thus, these results suggest that Nkx2-5 functions together with Xrn2 to regulate APA.

Xrn2 was recently reported to be recruited to TSSs and the downstream regions of genes (*Brannan et al., 2012*). Because Nkx2-5 was also localized at both regions, we next investigated whether Nkx2-5 recruited Xrn2 to genes for which the 3' UTR length was found to be regulated by Nkx2-5. Nkx2-5 knockdown reduced Xrn2 binding at the TSSs and downstream regions of genes but not at 3' UTR regions (*Figure 4E* and *Figure 4—figure supplement 2*). This result indicates a role for Nkx2-5 in Xrn2 binding to the TSSs and downstream regions of genes.

Next, we examined whether the knockdown of Nkx2-5 and Xrn2 generated extended poly(A)-tailed mRNAs. Northern blot analysis of *Tnnt2* and *Atp2a2* using poly(A)-tailed mRNA showed that the knockdown of Nkx2-5 and Xrn2 significantly increased the expression of extended poly(A)-tailed mRNAs (*Figures 4F,G*, and *Figure 4—figure supplement 3*). Long 3'UTRs have been reported to be related to decrease translation efficiency (*Mayr and Bartel, 2009*). Consistent with this, the amounts of Tnnt2 and Atp2a2 protein were slightly decreased in Nkx2-5- and Xrn2- knockdown eCMs (*Figure 4H* and *Figure 2—figure supplement 1B*). Next, we analyzed the increase in extended poly(A)-tailed mRNAs transcribed within 3 kb of the end of the 3' UTR of a reference sequence in Nkx2-5- and Xrn2-knockdown eCMs. Xrn2 knockdown also increased the expression of long 3' UTRs of genes for which the expression of long 3' UTRs was increased in Nkx2-5, but not Tbx5 nor Gata4, -knockdown eCMs (*Figure 4I* and *Figure 4—figure supplement 4*). Furthermore, Xrn2 knockdown strongly increased the expression of mRNAs at the transcription termination sites (TTSs), suggesting a role for Xrn2 in processing the ends of mRNAs. These results suggest that Xrn2 is involved in Nkx2-5-dependent regulation of APA.

## Interaction between Nkx2-5 and Xrn2 during heart development

To demonstrate the coordinated functions of Nkx2-5 and Xrn2 in heart development, we introduced siRNAs directed against Nkx2-5 and Xrn2 along with GFP expression plasmids into early-head-fold-stage embryos (E7.5, before the formation of the linear heart tube) (*Figures 5A and B*). These embryos were divided into three categories according to their heart looping morphologies (*Figure 5C*). The knockdown of both genes together significantly increased the number of abnormal hearts that remained in an essentially linear conformation compared with the knockdown of each gene alone (*Figure 5D*). We next examined whether the expression of genes related to Left-Right signaling pathway are affected by the injection of siRNAs against Nkx2-5 and Xrn2 (*Figures 5E and F*), since heart looping is known to be regulated by genes related to Left-Right signaling pathway (*Hamada et al., 2002*). Knockdown of Nkx2-5 and Xrn2 did not significantly change the expression of these genes including Pitx2, Nodal, Cryptic, and Lefty (*Figure 5E*). Moreover, Pitx2 expression profile was not affected by knockdown of Nkx2-5 and Xrn2 (*Figure 5F*). These results suggest that the coordinated functions of Nkx2-5 and Xrn2 are essential for heart formation.

Finally, we generated Xrn2-deleted mice by disrupting exons1 and 2 using the CRISPR/Cas9 system (*Figure 6* and *Figure 6—figure supplement 1*) (*Wang et al., 2013*). We deleted 11.6 kb from *Xrn2*, including exons1 and 2, which encode a part of the domain with enzymatic activity (*Figure 6—figure supplement 1*). To examine the genetic interaction between Nkx2-5 and Xrn2, we obtained *Nkx2-5^{+/-}Xrn2^{+/-}* newborns by intercrossing *Nkx2-5^{+/-}* mice with two *Xrn2^{+/-}* mice (#2 and #3). Although neither *Nkx2-5^{+/-}*nor *Xrn2^{+/-}* hearts showed a muscular ventricular septal defect (VSD) at

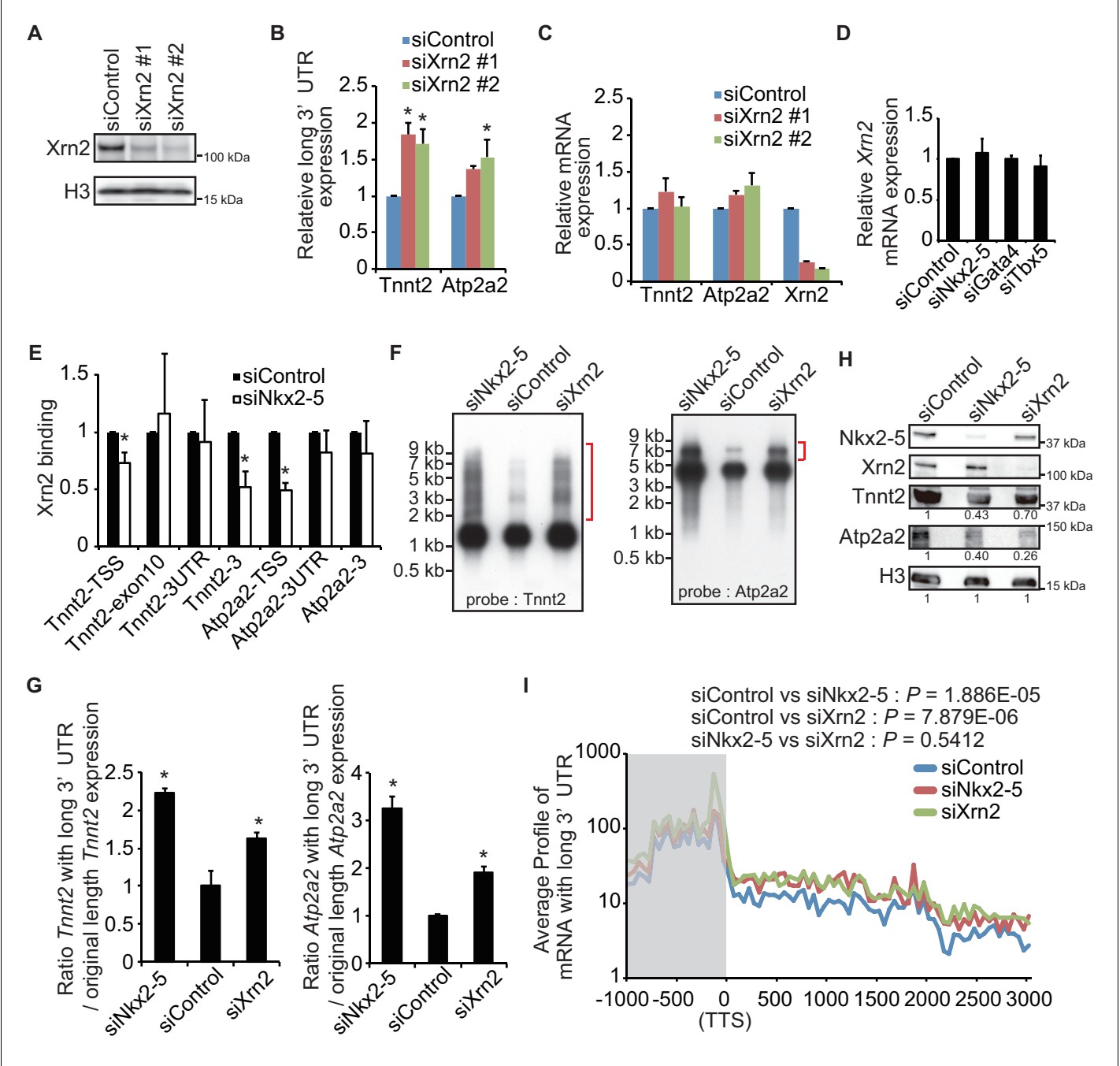

**Figure 4.** Nkx2-5 functions together with Xrn2 to regulate APA. (**A**) Xrn2 knockdown was analyzed by Western blotting. (**B** and **C**) qRT-PCR analysis of mRNAs expression of the long 3′ UTRs (**B**) and gene bodies (**C**) of *Tnnt2* and *Atp2a2* in Xrn2-knockdown eCMs, normalized to *Rplp2*. (**D**) *Xrn2* expression levels were measured by qRT-PCR and normalized to *Rplp2*. (**E**) Xrn2 binding in eCMs transfected with the indicated siRNAs was analyzed by ChIP-qPCR. The control values were set to 1.0. (**F**) Long 3′ UTRs in eCMs transfected with the indicated siRNA were analyzed by Northern blotting using probes against *Tnnt2* and *Atp2a2* mRNA. Red brackets indicate mRNAs with long 3′ UTRs. (**G**) The original lengths of the *Tnnt2* and *Atp2a2* mRNAs and the lengths of the *Tnnt2* and *Atp2a2* mRNAs with long 3′ UTRs that were used in the Northern blot analysis were measured by BAS5000. The ratio of the siControl was set to 1.0. (**H**) Tnnt2 and Atp2a2 proteins in Nkx2-5 and Xrn2 knockdown eCMs were analyzed by western blotting. (**I**) The average profiles of the mRNAs with long 3′ UTRs that were increased in Nkx2-5-knockdown eCMs are shown in eCMs transfected with the indicated siRNAs. The gray area indicates the coding region. Significance was assessed using the two-sample Kolmogorov-Smirnov test. For B, C, D, E, and G, error bars indicate the mean ± s.e.m. (*n* = 3). *, *p* < 0.05.

The following source data and figure supplements are available for figure 4:

*Figure 4 continued on next page*

*Figure 4 continued*

**Source data 1.** Source data for *Figure 4* and *Figure 4-figure supplement 1* and *2*.
**Figure supplement 1.** Knockdowns of Nkx2-5 or Xrn2 affect the expression of the long 3'UTR regions in chromatin-fractioned RNA.
**Figure supplement 2.** Knockdown of Nkx2-5 affects Xrn2-binding to *Myl7*.
**Figure supplement 3.** Nkx2-5 functions together with Xrn2 to regulate APA.
**Figure supplement 4.** Correlation analysis of Nkx2-5 binding and long 3'UTR expression.

postnatal day 0 (P0), we found a VSD in $Nkx2\text{-}5^{+/-}Xrn2^{+/-}$ newborn hearts (n = 3 of 7) (*Figure 6*). An atrial septal defect was observed in both $Xrn2^{+/-}$ and $Nkx2\text{-}5^{+/-}Xrn2^{+/-}$ mice, suggesting that Xrn2 contributes to atrial septum formation (*Figure 6*). These results indicate a genetic interaction between Nkx2-5 and Xrn2.

## Discussion

One of the critical roles of transcription factors is to regulate the expression level of target genes by recruiting transcription factor-associated factors such as chromatin remodeling factors, histone modifiers, and RNAPII to the promoter and enhancer regions of these genes (*Graf and Enver, 2009*). During heart development, the importance of cardiac transcription factors had been shown in many reports; for example, mutations in transcription factors, such as Nkx2-5, caused defects in heart development (*Bruneau, 2008*; *Srivastava, 2006*). However, the molecular mechanisms underlying the functions of cardiac transcription factors have remained unclear. Our results suggest that Nkx2–5 is involved in the 3'-end processing of target genes in conjunction with Xrn2; furthermore, Nkx2–5 and Xrn2 deficiency caused abnormalities in heart development. We identified the genome-wide occupancies of transcription factors and transcription factor-associated proteins and determined the histone modification signatures in E12.5 hearts. Active histone modifications, including H3K4me1/3, H3K36me3, and H3K27ac, were enriched in actively transcribed genomic regions, and repressive histone modifications, including H3K9me3 and H3K27me3, were enriched in silenced genes (*Figure 1* and *Figure 1—figure supplement 6*), which agrees with previous reports (*Barski et al., 2007*; *Wang et al., 2008*). While RNAPII-S5P has been reported to be associated with promoters (*Rahl et al., 2010*), we found enrichment of RNAPII-S5P at promoters and downstream regions of highly expressed genes in eCMs. This difference of RNAPII-S5P binding profile may be caused by differences of ChIP protocol and cell type. The transcription factor binding regions were different from those of exogenously tagged transcription factors in the murine adult cardiomyocyte cell line HL−1 (*He et al., 2011*). This difference may be caused by differences in gene expression during distinct developmental stages of cardiomyocytes. Metagene analysis revealed that Nkx2-5 is localized at both the TSSs and downstream regions of highly expressed genes in E12.5 hearts (*Figure 1* , *Figure 1—figure supplement 5*, and *Figure 1—figure supplement 6*). We compared ChIP and input data for obtaining peaks. Moreover, active histone modification ChIP-seq data including H3Keme1, H3K4me3, and H3K27ac showed the enrichment only around TSS regions (*Figure 1—figure supplement 7*). These indicate the enrichment of Nkx2–5 at the downstream regions is not bias in native ChIPseq experiment. Based on these data, Nkx2-5 plays unique roles in regulating the APA of highly expressed genes, including *Tnnt2* and *Atp2a2*, during heart development. The regulation of APA has been reported as one of the mechanisms for determining the length of the 3' UTR (*Elkon et al., 2013*). Long 3' UTRs are likely to decrease translation efficiency because they are recognized by miRNAs (*Elkon et al., 2013*). One function of Nkx2-5 may be to regulate the lengths of the 3' UTRs of genes involved in heart development through the regulation of APA, although there is a possibility that Nkx2-5 is also involved in the regulation of splicing. This process allows the production of necessary quantities of muscle proteins, such as cardiac troponin T (cTnT), a component of troponin that is essential for sarcomere assembly and is encoded by *Tnnt2* (*Nishii et al., 2008*), and *Atp2a2* (also known as SERCA2), a cardiac sarcoplasmic reticulum $Ca^{2+}$ pump that is essential for normal

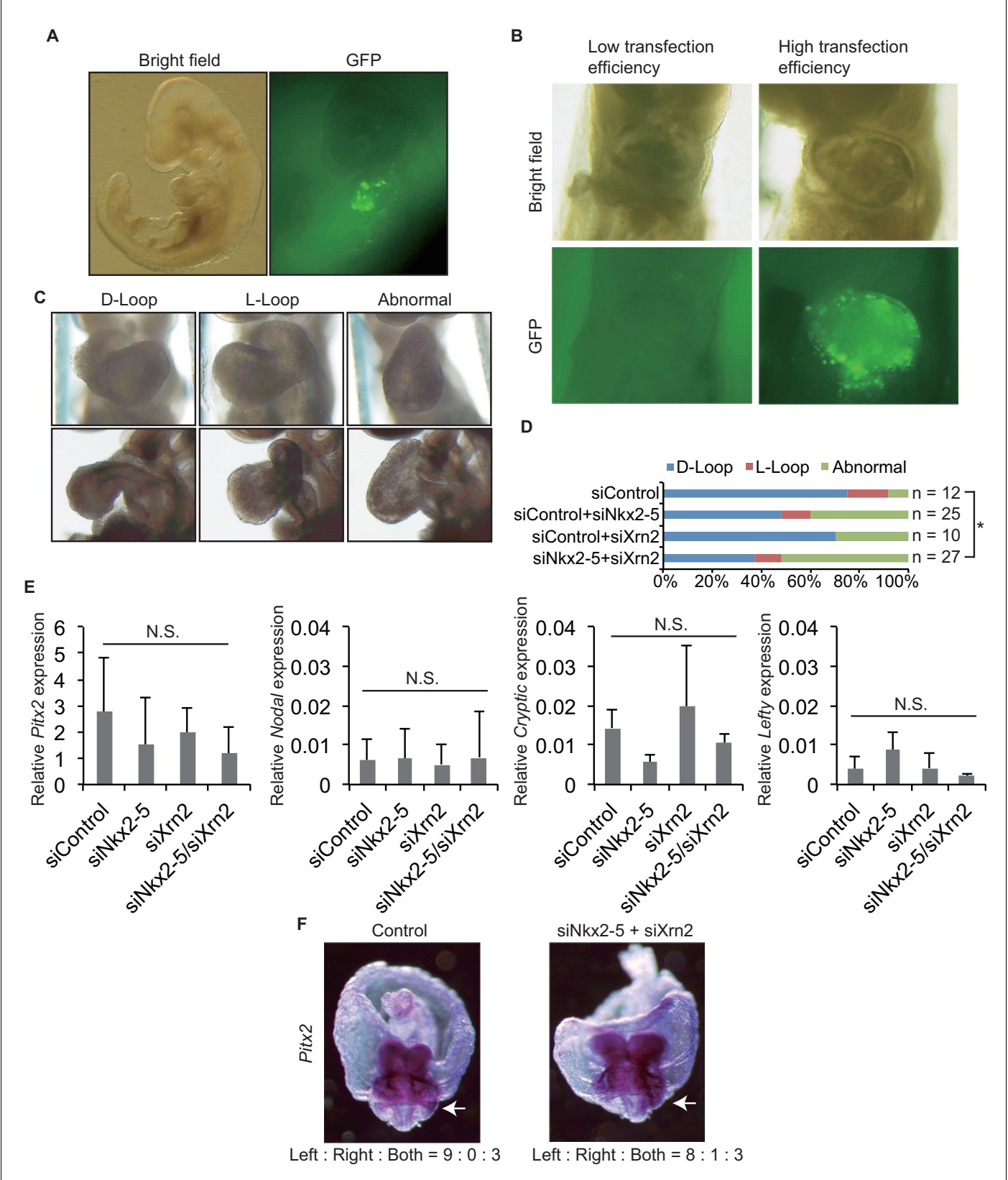

**Figure 5.** Knockdown of both Nkx2-5 and Xrn2 perturbs heart looping. (**A**) Transfection of siRNA into embryonic hearts. GFP was used to detect transfected fields. (**B**) We discarded the embryos with low transfection efficiency. (**C**) Representative morphologies of heart looping. D-Loop, the normal

*Figure 5 continued*

rightward loop; L-Loop, situs inversus; abnormal, hearts remained in an essentially linear conformation. (**D**) Knockdown of Nkx2–5 and Xrn2 in embryonic hearts. Graph bars indicate the% morphologies of heart looping. Significance was examined with Fisher's exact test. *, $p < 0.05$. (**E**) Looping-related genes expression level in siRNA-transfected embryonic hearts (n = 3), normalized to *Rplp2* expression level. n.s., not significant. (**F**) In situ hybridization of *Pitx2*. The numbers indicate *Pitx2* expression pattern among the right side, the left side, and the both sides. White arrows indicate lateral plate mesoderm.

cardiac performance (*Periasamy et al., 1999*). Moreover, Nkx2–5 has been reported to regulate the expression of miRNAs such as miR-1, which recognizes the 3' UTR of Cdc42 and decreases the Cdc42 protein level during heart development (*Qian et al., 2011*; *Zhao et al., 2007*). In addition, miRNAs might be involved in the regulation of translation efficiency by recognizing long 3' UTRs. Thus, APA dysregulation in Nkx2–5- and Xrn2-depleted hearts might cause abnormalities in heart development by decreasing the level of heart-related proteins such as Tnnt2 and Atp2a2. Our results suggest that in addition to the regulation of gene expression by Nkx2–5, the regulation of APA by Nkx2-5 and Xrn2 is an important mechanism for the precise progression of heart development.

Recent studies have reported significant interactions between chromatin regions in the nucleus and have shown the importance of interactions between promoters and enhancers for the activation of gene expression (*Kagey et al., 2010*; *Li et al., 2012*; *Wang et al., 2011*). Although the function of enhancer-promoter DNA looping mediated by transcription factors and mediators has previously been demonstrated (*Kagey et al., 2010*; *Wang et al., 2011*), genome-wide chromatin conformational analyses have revealed that (1) several chromatin regions form DNA loops between the enhancer and promoter and between the promoter and downstream regions of the gene (promoter-downstream looping) and (2) these interactions are associated with RNAPII (*Li et al., 2012*). Promoter-downstream looping has been shown to enhance RNAPII recycling from the TTS to the promoter (*Cavalli and Misteli, 2013*) and to determine the direction of transcription (*Tan-Wong et al., 2012*). In this study, we showed that Nkx2-5 deficiency increased the expression of long 3' UTRs beyond the TTS of the studied genes, which occurred concomitantly with changes in the stability of promoter-downstream looping and the amount of RNAPII localized downstream of the genes

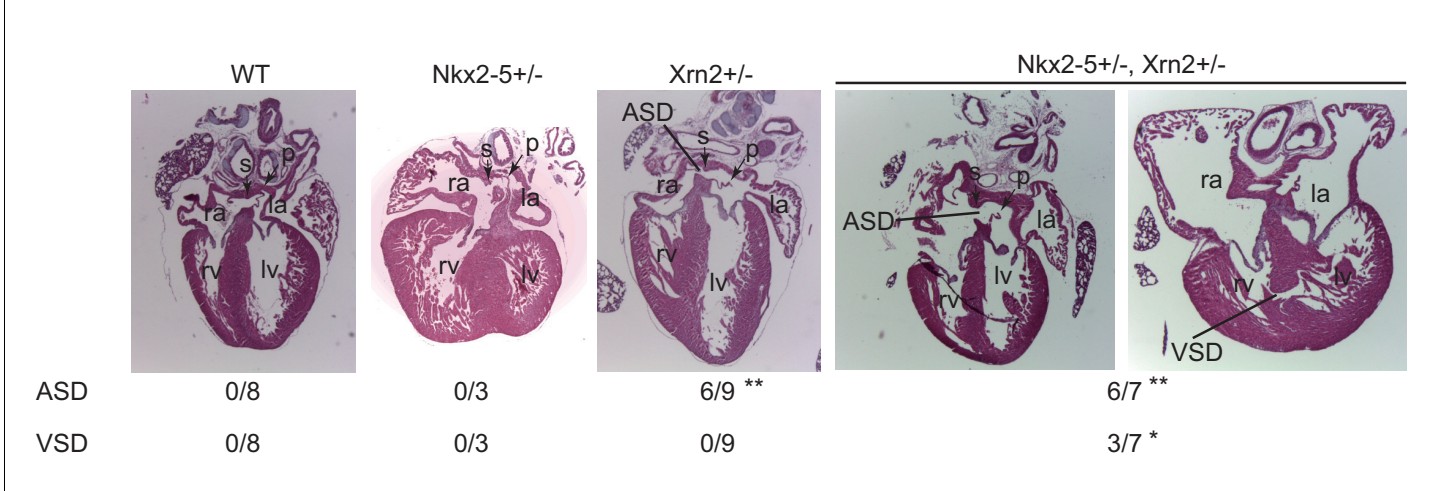

**Figure 6.** Nkx2-5 genetically interacts with Xrn2. Histological analysis of *Nkx2-5*[+/-] and *Xrn2*[+/-] newborn hearts. Frontal sections from newborn hearts were stained with hematoxylin and eosin. ASD was observed in *Xrn2*[+/-] (n = 6 of 9) and *Nkx2-5*[+/-]*Xrn2*[+/-] (n = 6 of 7) newborns. VSD was observed in *Nkx2-5*[+/-]*Xrn2*[+/-] newborns (n = 3 of 7). lv, left ventricle; rv, right ventricle; la, left atrium; ra, right atrium; p, septum primum; s, septum secundum. *, $p < 0.05$. **, $p < 0.01$.

The following figure supplement is available for figure 6:

**Figure supplement 1.** *Nkx2-5 genetically interacts with Xrn2.*

(*Figure 2*). Thus, promoter-downstream looping is likely to be involved not only in recycling RNAPII (*Cavalli and Misteli, 2013*) but also in regulating APA.

Xrn2 is known to have 5'-3' exonuclease activity and to contribute to transcription termination at the ends of genes (*Kim et al., 2004*; *West et al., 2004*). Recently, it was reported that Xrn2 along with mRNA decapping factors bound to TSSs and other downstream regions and was involved in the regulation of RNAPII elongation (*Brannan et al., 2012*). Our results suggest that Nkx2-5 is involved in the regulation of Xrn2 binding to both TSSs and the downstream regions of target genes during heart development and that promoter-downstream looping support the function of Xrn2 at these regions (*Figures 2* and *4*). The association of Nkx2−5 with Xrn2 was EtBr- and RNaseA-resistant (*Figure 3*), and the Nkx2−5 HD was not required for the association of Nkx2−5 with Xrn2 (*Figure 3*). These findings suggest that Nkx2−5 may be able to recruit Xrn2 to target genomic regions that Nkx2-5 recognized via its HD. Although Xrn2 is a 5'-3' exonuclease and is believed to eliminate 3' cleavage products that remain associated with RNAPII after mRNA cleavage (*Kuehner et al., 2011*; *Richard and Manley, 2009*), Xrn2 knockdown increased the expression of mRNAs with long 3' UTRs, suggesting that Xrn2 is also critical for regulating APA and promotes the usage of proximal poly(A) sites. Nkx2−5 functions together with Xrn2 but not with RNA helicases, which are also transcription termination factors; therefore, both Xrn2 and transcription factors might be involved in determining the usage of poly(A) sites.

Recent deep-sequencing analyses have revealed that over 70% of human genes are estimated to have 3' UTRs of variable lengths and that these differences are dependent on cell type and are regulated by APA (*Elkon et al., 2013*). APA is modulated by the processing of mRNA at proximal or distal poly(A) sites. mRNAs with short 3' UTRs are generated in highly proliferative cells including cancer cells (*Mayr and Bartel, 2009*), and these mRNAs lack miRNA recognition regions; thus, their translation efficiency can be increased by avoiding miRNA (*Mayr and Bartel, 2009*). Nerve cells are known to generate mRNAs with long 3' UTRs by utilizing distal poly(A) sites (*Miura et al., 2013*). The termination of transcription by RNAPII requires RNA 3'-end processing and termination factors (*Kuehner et al., 2011*), and defects in these factors increase the level of RNA with long 3'UTR due to the dysregulation of RNAPII termination (*Liu et al., 2012*; *Padmanabhan et al., 2012*). Although cleavage and polyadenylation factors including cleavage and polyadenylation specificity factor (CPSF), cleavage-stimulating factor (CSTF), and cleavage factor Im (CFIm), regulate APA (*Elkon et al., 2013*; *Masamha et al., 2014*), the mechanisms that regulate 3' UTR length remain undefined. Our findings suggest that Nkx2−5, an essential cardiac transcription factor, regulates APA through the recruitment of 5'-3' exonuclease Xrn2 during heart development (*Figure 4*) and that tissue-specific transcription factors play an important role in tissue-specific 3' UTR length.

Although knockdown of Nkx2-5 and Xrn2 showed the defect of heart looping (*Figure 5*), Nkx2-5 and Xrn2 double heterozygotes have ASD and VSD (*Figure 6*). This difference may be caused by the remaining amount of protein expression and / or acute decrease of expression by gene knockdowns or stable decrease of expression by gene knockout, which is supported by the recent study reporting the difference of phenotypes caused by gene knockdowns or gene knockouts (*Rossi et al., 2015*). Consistent with our results, Nkx2-5 heterozygotes do not display defective heart looping, although Nkx2-5 homozygotes show this defect (*Biben et al., 2000*).

In summary, our findings suggest that Nkx2−5 regulates APA by recruiting Xrn2 to targeted genomic regions. Deficiencies in Nkx2−5 and Xrn2 disrupted the regulation of 3' UTR length and resulted in abnormalities in heart formation. Although we cannot exclude the possibility of other defects in addition to the dysregulation of APA, the data suggest that APA dysregulation could be one of the mechanisms that cause CHD in patients with mutated Nkx2-5, nevertheless, it is still unclear whether long 3'UTR per se is related to the pathogenesis of cardiac abnormality. Although further studies are required to elucidate the molecular role of Nkx2-5 in the regulation of APA, our findings provide a conceptual framework for understanding how transcription factors regulate 3' UTR length.

## Materials and methods

### Mice

Embryonic hearts were obtained from C57BL/6 wild-type and *Nkx2-5*-deficient mice (*Moses et al., 2001*) in accordance with protocols 3422–1 and 24-084-012 approved by the Ethics Committee for Animal Experiments of the Osaka University Graduate School of Medicine.

### Generation of Xrn2-deficient mice using CRISPR/Cas9

Cas9 RNA and gRNA were generated by in vitro transcription with the SP6 mMessage mMachine kit (Ambion, Foster City, CA) as previously described, with minor modifications (*Wang et al., 2013*). Cas9 RNA and two gRNAs were injected into fertilized eggs obtained from C57BL/6 mice to delete exon 1 and 2 of *Xrn2*. Deletion of these exons was confirmed by sequencing. *Xrn2* heterozygous mice were obtained by intercrossing Xrn2 F0 with C57BL/6 mice.

### Native ChIP-seq

Nuclear extracts from four E12.5 embryonic hearts were used in each native ChIP experiment following a previously described protocol with minor modifications (*Nimura et al., 2006*; *2009*). Isolated nuclei from embryonic hearts were treated at 25°C for 30 min with 4.8 U ml$^{-1}$ micrococcal nuclease in 250 µl of a nuclear isolation buffer containing 400 mM NaCl, which was then diluted to 200 mM NaCl. The digested chromatin was immunoprecipitated with 15–50 µg of antibody. Only mono- and di-nucleosomes size DNA was used for construction of sequencing libraries. Sequencing libraries were prepared from two or more biological-replicate ChIP samples and from an input according to the instructions provided with the SOLiD Fragment Library Barcoding Kit (Life Technologies, Carlsbad, CA). The libraries were sequenced with SOLiD 4. The resulting reads were mapped using BioScope software (Life Technologies) with the default configuration, combined biological replicates, and analyzed using Homer (*Heinz et al., 2010*), CEAS (*Shin et al., 2009*) and R software programs. The mapping results are shown in *Supplementary file 1A* and were generated using IGV software (*Robinson et al., 2011*). The heatmap was generated using Java TreeView (http://jtreeview.sourceforge.net/).

### RNA-seq

RNA was extracted from primary embryonic cardiomyocytes (from E12.5 hearts) and E9.5 hearts from *Nkx2-5*-deficient/wild-type littermates using the TRIzol reagent (Invitrogen, Carlsbad, CA) according to the manufacturer's instructions. Strand-specific sequencing libraries from two biological replicate RNA samples were prepared according to the Life Technologies protocol as previously described (*Mori et al., 2012*). Briefly, digested poly(A)-tailed mRNA was ligated to the SOLiD Adaptor Mix and then reverse-transcribed using the SOLiD Total RNA-Seq Kit (Life Technologies). Size-selected first-strand cDNA was amplified by using SOLiD 5' PCR primers and barcoded SOLiD 3' PCR primers (Life Technologies). RNA-seq libraries were sequenced with SOLiD 4. The resulting reads were mapped using BioScope software (Life Technologies) and analyzed using BioScope and Cufflinks (*Trapnell et al., 2010*; *2012*). The mapping results are shown in *Supplementary file 1B*.

RefSeq genes were divided into three categories (High, Middle, and Low) according to the FPKM (fragments per kilobase of exon per million fragments mapped) of each gene in wild-type eCMs: High: FPKM > 500; Middle: FPKM 100–500; and Low: FPKM 10–100. The genes identified for analysis were either upregulated to more than a ln (fold change) of 0.2 or downregulated to less than a ln (fold change) of 0.2 compared with control siRNA-treated eCMs. The High, Middle, and Low categories included 90, 717, and 5516 genes, respectively.

The expression of read-through RNA (FPKM > 0.5) was measured within 5 kb in eCMs and in Nkx2-5-deficient embryonic hearts using BioScope (Life Technologies). Genes with an average change in read-through RNA expression of more than a 1.4-fold compared with control siRNA-treated eCMs or wild-type littermate hearts were analyzed using DAVID. Genes for which the read-through RNA was expressed at less than 0.5-fold compared with controls were considered to have downregulated read-through RNA. Genes with expression within regions of a read-through RNA were removed from the analysis. The normalized read-through RNA tags were counted in 50-bp bins

using CEAS, and the obtained values were then used to calculate the average expression in each bin.

The polyadenylation sites were obtained from APADB (http://tools.genxpro.net/apadb/).

## Purification of chromatin-fractioned RNA from eCMs

Chromatin-fractioned RNA was purified from 2~5 X $10^6$ cells of eCMs as previously described (*Nojima et al., 2015*). rRNA was depleted using Ribominus Eukaryote kit ver2 (Life technologies) from 3.5~4.0 µg chromatin RNA.

## Antibodies

The antibodies used in this study are shown in *Supplementary file 1D*. All the antibodies used for ChIP-seq have been previously reported as suitable or were verified to be suitable for the immuno-precipitation of target proteins (*Figure 1—figure supplement 4*).

## Knockdown in cardiomyocytes

Primary eCMs were obtained from E12.5 hearts by overnight digestion with trypsin-EDTA at 4°C and separation from cardiac fibroblasts by pre-plating on collagen type I-coated dishes for 1 hr, as previously described (*Springhorn and Claycomb, 1989*) with some modifications. The cardiomyocytes were transfected with Nkx2-5 (Mm_Nkx2-5_8958, SIGMA), Tb x 5 (Mm_Tb x 5_9160, SIGMA), Gata4 (Mm_Gata4_5094, SIGMA), Xrn2 (Mm_Xrn2_3520 and Mm_Xrn2_3522, SIGMA), or control (SIC-001, SIGMA) siRNAs using RNAiMAX (Invitrogen) and cultured for 48 hr in Dulbecco's modified Eagle medium supplemented with 10% fetal bovine serum. The siRNAs directed against Nkx2-5, Tbx5, and Gata4 were chosen from three different siRNAs after determining the knockdown efficiency and specificity of each candidate using quantitative RT-PCR and Western blotting.

## Quantitative RT-PCR

Total RNA was extracted with the TRIzol reagent (Invitrogen). Reverse transcription was performed with SuperScript III (Invitrogen) as previously described (*Nimura et al., 2009*) and analyzed using the CFX384 Real-Time System (BIO-RAD, Hercules, CA). Genomic DNA contamination was evaluated by examining reverse transcription reaction samples lacking reverse transcriptase. The values were normalized to Rplp2 (ribosomal protein, large, P2) and expressed relative to the values obtained with control siRNA-treated eCMs.

## Native 3C

Nuclei isolated from $10^6$ siRNA-transfected E12.5 primary cardiomyocytes were digested overnight at 37°C with *Eco*RI, *Bgl*II, or *Hin*dIII (100 units per $10^6$ cells, TOYOBO, JAPAN) in 100 µl of buffer with 0.4% NP-40 and complete EDTA-free protease inhibitors (Roche, Indianapolis, IN). The nuclei were ligated overnight with Ligation High (TOYOBO) and then extracted using phenol-chloroform. The primers were designed using the 3C Primer (http://dostielab.biochem.mcgill.ca/index.php) and Primer3 programs (http://frodo.wi.mit.edu/). BAC plasmids RP23-2E23 for Tnnt2 and RP23-128I8 for Atp2a2 were used as controls. PCR was performed using the THUNDERBIRD SYBR qPCR mix (TOYOBO) and a CFX384 thermocycler (BIO-RAD). The amplicons were separated on 2% agarose gels stained with ethidium bromide for visualization.

## Northern blotting

Poly A-tailed mRNA was purified from total RNA that was extracted using TRIzol (Invitrogen) and the Ambion MicroPoly(A) Purist Kit (Ambion). The poly(A)-tailed mRNA (360 ng) was separated by 1% agarose gel electrophoresis and transferred to a Hybond-N+ nylon transfer membrane (Amersham Biosciences, Pittsburgh, PA). *Tnnt2* and *Atp2a2* mRNA were detected with $^{32}$P-labeled cDNA that contained a portion of the gene-body regions of *Tnnt2* and *Atp2a2*. The intensities of the bands were quantitatively measured using a BAS5000 (GE Healthcare, Pittsburgh, PA).

## Immunoprecipitation in C3H10T1/2 cells

Expression constructs containing HA-Nkx2-5, Flag-Xrn2, and their deletion mutants were cotransfected into C3H10T1/2 cells, purchased from ATCC, in 10 cm dishes as previously described

(**Nimura et al., 2009**). This cell was neither authenticated nor tested for mycoplasma contamination. HA-Nkx2-5, Flag-Xrn2, and their respective deletion mutants were immunoprecipitated from nuclear extracts that were prepared as previously described (**Nimura et al., 2009**). Co-immunoprecipitated proteins were analyzed by Western blotting.

### Introduction of siRNA into embryonic hearts

E7.5 embryos were dissected from the uterus, and embryos at the early-head-fold stage were carefully selected. Liposomes were created by mixing 3.33 µM siRNAs directed against Nkx2-5, Xrn2, or control together with 53 ng/µl EGFP expression vector in 15 µl of OPTI-MEM with 3 µl of RNAiMAX (Invitrogen) diluted in 15 µl of OPTI-MEM. Liposomes were injected into heart fields as previously described (**Yamamoto et al., 2004**). The embryos were rotationally cultured for 48 hr in Dulbecco's modified Eagle's medium supplemented with 75% rat serum. GFP signals and heart morphology were examined with a Leica M165FC microscope, and the embryos with weak GFP signals were discarded (**Figure 5**).

### In situ hybridization

Whole-mount in situ hybridization was performed according to standard procedures (**Wilkinson, 1992**) and probe specific for Pitx2 mRNA (**Yoshioka et al., 1998**).

### Statistical analysis

The data are presented as the mean ± s.e.m. $P$ values were calculated using the two-tailed $t$-test and the Tukey-Kramer HSD test using JMP, Fisher's exact test was performed using http://aoki2.si.gunma-u.ac.jp/exact/fisher/getpar.html, and the two-sample Kolmogorov-Smirnov test was performed using R. $p < 0.05$ was considered statistically significant. The number of $n$ shows biological replication.

## Acknowledgements

We thank M Okado and T Hiraoka for technical assistance, T Shimbo for critical discussion, S Tsutsumi for discussion of the ChIP-seq data analysis, T Ishikura and K Azuma for SOLiD sequencing, and members of the GTS laboratory for discussion and support. This study was supported by MEXT KAKENHI Grant Nos. 23710224, 24116514, and 26116718 and grants from the Japan Heart Foundation, the Kanae Foundation for the Promotion of Medical Science, the Takeda Science Foundation, and the Osaka University Program for the Support of Networking among Present and Future Researchers to KN.

## Additional information

### Funding

| Funder | Grant reference number | Author |
| --- | --- | --- |
| MEXT KAKENHI | 23710224 | Keisuke Nimura |
| Japan Heart Foundation | | Keisuke Nimura |
| Kanae Foundation for the Promotion of Medical Science | | Keisuke Nimura |
| Takeda Science Foundation | | Keisuke Nimura |
| Osaka University | | Keisuke Nimura |
| MEXT KAKENHI | 24116514 | Keisuke Nimura |
| MEXT KAKENHI | 26116718 | Keisuke Nimura |

The funders had no role in study design, data collection and interpretation, or the decision to submit the work for publication.

## Author contributions
KN, Designed and performed the experiments, Analyzed the data, Wrote the manuscript, Contributed unpublished essential data or reagents; MY, Performed the experiments, Analyzed the data, Drafting or revising the article; MT, KS, Performed the experiments, Analyzed the data; KT, Generated the Xrn2-deleted mice, Acquisition of data, Analysis and interpretation of data; NK, HNi, HNa, Acquisition of data, Analysis and interpretation of data; SI, Acquisition of data; TT, Performed the experiments; RJS, Contributed mouse resources, Acquisition of data; HA, Contributed sequencing, Acquisition of data; YK, Provided support and general guidance for this study, Wrote the manuscript, Analysis and interpretation of data

## Author ORCIDs
Keisuke Nimura, http://orcid.org/0000-0001-9680-2646

## Ethics
Animal experimentation: The protocols (3422-1 and 24-084-012) are approved by the Ethics Committee for Animal Experiments of the Osaka University Graduate School of Medicine.with protocols.

# Additional files

## Supplementary files
• Supplementary file 1. Lists of mapping results, primers, and antibodies. (A) Mapping results of ChIP-seq . (B) Mapping results of RNA-seq. (C) Primers used in this study. (D) Antibodies used in this study.

## Major datasets
The following dataset was generated:

| Author(s) | Year | Dataset title | Dataset URL | Database, license, and accessibility information |
|---|---|---|---|---|
| Keisuke Nimura | 2016 | Transcription regulation during heart development | http://www.ncbi.nlm.nih.gov/sra/?term=DRA002229 | Publicly available at the NCBI Short Read Archive (accession no: DRA002229) |

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
