## [Decision Letter]

Thank you for submitting your article "Regulation of alternative polyadenylation by Nkx2-5 and Xrn2 during heart development" for consideration by *eLife*. Your article has been favorably evaluated by James Manley as the Senior editor and three reviewers, including Maurice Swanson (Reviewer #3) and Nick Proudfoot (Reviewer #1), who is a member of our Board of Reviewing Editors.

The reviewers have discussed the reviews with one another and the Reviewing Editor has drafted this decision to help you prepare a revised submission.

This paper describes an interesting data set showing that the cardiac development associated transcription factor Nkx2-5 plays an important role in regulating mRNA 3' end formation for a number of more highly expressed cardiac genes.

I am pleased to tell you that all three consider the paper interesting and worthy of publication provided fairly extensive additional data can be added to strengthen the paper. I list below the key points that need to be addressed in revision.

1) The effect of Nkx2-5 or Xrn2 knock down produces apparently only relatively small effects on alternative polyA site (APA) selection (1.5 to 2 fold). However, this data is based on total RNA-seq analysis so will not directly measure the selective use of PAS as it will also be affected by RNA stability differences between mRNA with different 3'UTRs (see Neve et al. Gen Res 26, 24-35 2016). More convincing APA data would come from for instance RNA-seq on chromatin fractionated RNA to look at nascent transcripts. This type of data would also allow them to determine if Nkx2-5 plays a role in regulating alternative splicing.

2) The evidence that Nkx2-5 is required for Xrn2 recruitment to gene 3' ends based on ChIP is a bit marginal. More primer pairs are needed across *Tnnt2* as well as the analysis of additional Nkx2-5 sensitive genes. Also a RIP approach might produce more or complementary convincing data.

3) The 3C analysis on *Tnnt2* and *Atp2a2* needs more primers across the gene to convincingly show a promoter to PAS region interaction that is specifically sensitive to loss of Nkx2-5. What are the control lanes in Figure 2?

4) The manuscript’s text and figures need improved clarity and clearer annotation. pA site usage and transcription termination should be distinguished. The positions of annotated PAS should be clearly shown on all figures

5) A summary figure of interacting regions between Nkx2-5 and Xrn2 would be helpful in Figure 3.

6) Figure 4 needs internal/loading controls for the Northern blots.

---

## [Author Response]

1) The effect of Nkx2-5 or Xrn2 knock down produces apparently only relatively small effects on alternative polyA site (APA) selection (1.5 to 2 fold). However, this data is based on total RNA-seq analysis so will not directly measure the selective use of PAS as it will also be affected by RNA stability differences between mRNA with different 3'UTRs (see Neve et al. Gen Res 26, 24-35 2016). More convincing APA data would come from for instance RNA-seq on chromatin fractionated RNA to look at nascent transcripts. This type of data would also allow them to determine if Nkx2-5 plays a role in regulating alternative splicing.

According to your kind suggestions, we examined the expression of long 3’UTR of *Tnnt2* and *Atp2a2* on chromatin-fractioned RNA by RNA-seq and qPCR (Figure 4—figure supplement 1). RNA-seq clearly showed the increase in expression of long 3’UTR of *Tnnt2* and *Atp2a2* (Figure 4—figure supplement 1). The expression of these regions was comparable to the expression of long 3’UTR of mRNAs in total RNA (Figure 4 and Figure 4—figure supplement 1). RNA-seq of Nkx2-5 knockout hearts also showed distinct difference of the usage of poly(A) sites, compared to wild-type hearts (Figure 1—figure supplement 9). These results suggest that Nkx2-5 is involved in the regulation of alternative polyadenylation.

2) The evidence that Nkx2-5 is required for Xrn2 recruitment to gene 3' ends based on ChIP is a bit marginal. More primer pairs are needed across Tnnt2 as well as the analysis of additional Nkx2-5 sensitive genes. Also a RIP approach might produce more or complementary convincing data.

Although the Xrn2-binding at the TSS and long 3’UTR region of *Tnnt2* was decreased in Nkx2-5-knockdown eCMs, we examined the primer for amplifying the exon 10 of *Tnnt2* in Figure 4. Xrn2-binding at this region was not affected by Nkx2-5 knockdown (Figure 4). *Myl7* gene is one of Nkx2-5 sensitive genes (Figure 1—figure supplement 8), and was used for further confirmation of the effect of Nkx2-5-knockdown for Xrn2-binding. Xrn2-binding was also decreased at the long 3’UTR region of *Myl7* in Nkx2-5-knockdown eCMs (Figure 4—figure supplement 2). These results show that Nkx2-5 is required for Xrn2 recruitment to the downstream regions of the target genes.

We tried RNA-immunoprecipitation analysis of Xrn2. Although Nkx2-5 might affect the association between Xrn2 and the long 3’UTR regions at *Tnnt2* and *Atp2a2* mRNAs (see Figure 7), the efficiency of precipitation of RNA at the long 3’UTR regions by Xrn2 was a little low. As one of the reasons, it might be conceivable that the nuclease activity of the complex including Xrn2 might be high at the poly-adenylation sites, meaning that the mRNAs might be quickly released from the complex including Xrn2. Also, it is quite difficult to increase cell numbers as much as used for general RIP analysis, more than 1X10E7, since we used primary cardiomyocytes obtained from E12.5 hearts. Thus, we thought that this result might not be conclusive for demonstrating the effect of Nkx2-5 on the interaction between Xrn2 and the target mRNAs.

Author response image 1.RNA-immunoprecipitation analysis of Xrn2 in eCMs.Relative Xm2-binding to mRNA at the downstream regions (-3) of *Tnnt2* (18.4 kb) and *Atp2a2* (47.2 kb)was analyzed by RIP in eCMs transfected the indicated siRNAs. The mRNA of long 3’ UTRs of *Tnnt2* and *Atp2a2* values were normalized to input. The control values were set to 1.0. Error bars indicate the means ± s.e.m. (n = 3). *, p<0.05.**DOI:**
http://dx.doi.org/10.7554/eLife.16030.031

3) The 3C analysis on Tnnt2 and Atp2a2 needs more primers across the gene to convincingly show a promoter to PAS region interaction that is specifically sensitive to loss of Nkx2-5. What are the control lanes in Figure 2?

In Figure 2, we added the primer for 14.0 kb region from the promoter at *Tnnt2* gene and the primer for 32.3 kb from the promoter at *Atp2a2* gene. The results show that knockdowns of Nkx2-5, Tbx5, and Gata4 do not affect the interaction between the promoter and 14.0 kb region at *Tnnt2* gene and the promoter and 32.3 kb region at *Atp2a2* gene. At the controls in Figure 2, the primers for amplifying the undigested regions by restriction enzymes were used to show input DNA as control.

4) The manuscript’s text and figures need improved clarity and clearer annotation. pA site usage and transcription termination should be distinguished. The positions of annotated PAS should be clearly shown on all figures

We put the positions of pA sites on Figure 1, Figure 1—figure supplement 5, Figure 1—figure supplement 8, Figure 1—figure supplement 9, Figure 2, and Figure 4—figure supplement 1. We could not obtain the positions of pA sites in *Tnni1* gene. We changed the description as per your kind suggestions.

“Our results indicate that Nkx2-5 regulates not only the initiation but also the usage of poly(A) sites during heart development.”

“and defects in these factors increase the level of RNA with long 3’UTR due to the dysregulation of RNAPII termination (Liu et al., 2012; Padmanabhan et al., 2012).”

5) A summary figure of interacting regions between Nkx2-5 and Xrn2 would be helpful in Figure 3.

Thank you for the kind suggestion. We added schematic presentation of deletion mutants and summary of interacting regions between Nkx2-5 and Xrn2 in Figure 3 to easily understand the results.

*6) Figure 4 needs internal/loading controls for the Northern blots.*

We added the EtBr-staining gels and northern blots of b-actin as loading controls for the Northern blots at Figure 4 in Figure 4—figure supplement 3.